# SQT – ROUGH CONSERVATIVE ACTOR CRITIC

## ABSTRACT

Std $Q$-target is a conservative actor critic ensemble based $Q$-learning algorithm which based on a single key $Q$-formula–$Q$-networks standard deviation, an uncertainty penalty. A minimalistic solution to the problem of overestimation bias. We implement SQT on top of actor critic and test it against the SOTA actor critic algorithms on popular MuJoCo tasks. SQT shows a clear performance advantage over TD3, SAC and TD7 on the tested tasks majority.

## 1 INTRODUCTION

RL is the problem of finding an optimal policy, $\pi : \mathcal{S} \to \mathcal{A}$, mapping states to actions, by an agent which makes a decision in an environment and learn by trial and error. We model the problem by an MDP Puterman (1994). Let $a_t \in \mathcal{A}$ be an action chosen at timestep $t$ at state $s_t \in \mathcal{S}$, leads to the state $s_{t+1} \in \mathcal{S}$ in the probability of $P(s_{t+1}|s_t, a)$ and resulting in the immediate reward of $r_{t+1}$.

$Q$-learning Watkins & Dayan (1992), introduced in Watkins' thesis Watkins (1989), is a widely-used tabular model-free algorithm that suffers from overestimation bias. This bias occurs when the $Q$-values for a policy $\pi$ are overestimated relative to the true $Q$-values. The source of this issue is the optimization of $Q$-values with respect to the argmax policy, which selects the maximum $Q$-value during updates. These overestimated values are then **propagated** throughout the entire table via the update process, ultimately degrading the algorithms performance.

Double $Q$-learning Hasselt (2010) solves this problem by introducing the **double estimator**, updating $Q^A$ by $Q^B$ $Q$-values with the action of $a^* = \arg \max_{a \in \mathcal{A}} Q^A$, ensures that $a^*$ is unbiased w.r.t. $Q^B$, turning the problem into an **underestimation bias**–where the $Q$-estimations are lower than the real MC returns–which cause to a poor exploration and finally to poor performance.

Weighted Double $Q$-learning Zhang et al. (2017) seeks to strike a balance between traditional $Q$-learning and Double $Q$-learning by introducing a weighting factor $w \in (0, 1)$. This factor combines the single-estimator and double-estimator in a unified update rule, with the $Q$-target estimation given by:

$$y_{\text{WDQL}} = r + \gamma[w \cdot Q^A(s', a^*) + (1 - w) \cdot Q^B(s', a^*)], \quad a^* = \arg \max_{a \in \mathcal{A}} Q^A(s', a) \qquad (1)$$

where the weighting factor $w$ is updated as:

$$w \leftarrow \frac{|Q^B(s, a^*) - Q^B(s, a_L)|}{c + |Q^B(s, a^*) - Q^B(s, a_L)|} \qquad (2)$$

This formulation adjusts $Q^A$'s weight based on the action gap of $Q^B$, providing a per-$Q$-network uncertainty measure that emphasizes $Q^A$ when the action gap in $Q^B$ is larger.

In this paper, we introduce a different approach to tackle the problem of overestimation bias, with a minimal coding effort, using an ensemble based $Q$-networks disagreement, that serves as a **penalty for uncertainty**, in the heart of the $Q$-target estimations formula.

## 2 BACKGROUND

SQT leverages ensemble methods as a mechanism to address overestimation bias, combining mathematical rigor with psychological principles. Mathematically, SQT minimizes the TD error through critic updates using an ensemble of $Q$-networks. The loss function is formulated as:

$$L(\theta^Q) = \mathbb{E}_{s_t \sim \rho^\beta, a_t \sim \beta, r_t \sim E} \left[ \left( Q_i(s_t, a_t | \theta^Q) - y_t \right)^2 \right], \tag{3}$$

where the target $y_t$ is defined by:

$$y_t = r(s_t, a_t) + \gamma \mathcal{Q}_{i=1\ldots N}[Q](s_{t+1}, \mu(s_{t+1}) | \theta^Q) - \alpha \cdot \text{SQT}[\mathcal{B}]. \tag{4}$$

In this formulation, the ensemble operator $\mathcal{Q}_{i=1\ldots N}$ combines predictions from $N$ distinct $Q$-networks to compute the $Q$-target, while the term $\text{SQT}[\mathcal{B}]$, defined as the standard deviation of $Q$-network outputs, serves as a regularization mechanism. This regularization penalizes excessive disagreement between networks, which can result from overestimations. The use of a batch-mean standard deviation ensures that this penalization adapts dynamically based on the uncertainty in the predictions.

From a psychological perspective, this approach mirrors how humans make decisions under uncertainty. In situations where biases or noise may distort judgment, individuals often consult multiple perspectives or experts to reduce the impact of a single, potentially biased viewpoint. This concept, known as the wisdom of the crowd, is reflected in SQT's ensemble method, where multiple $Q$-networks provide diverse estimates, resulting in a more balanced and accurate policy. The diversity of network outputs serves to offset optimistic biases that can arise from relying on a single estimator.

This mathematical strategy of aggregating ensemble predictions to compute target values, combined with a penalty on disagreement, captures the cognitive process of weighing diverse opinions to reduce bias and make more reliable decisions. In human psychology, consulting multiple perspectives leads to a reduction in overconfidence and improved decision accuracy. Similarly, in SQT, the ensemble method mitigates overestimation bias by curbing the excessive optimism that can emerge from noise or errors in individual $Q$-networks.

By uniting these mathematical and psychological insights, ensemble methods offer a powerful solution to overestimation bias in reinforcement learning. They provide both a formal mechanism for smoothing out optimistic value predictions and a cognitive framework for incorporating diverse estimates, promoting more cautious and reliable decision-making Tversky & Kahneman (1974); Surowiecki (2005).

## 3 BIAS

Overestimation bias, a cognitive phenomenon rooted in human psychology, manifests as the tendency to overestimate abilities, outcomes, or success probabilities in uncertain contexts. In behavioral economics and psychology, this bias is exemplified by the Dunning-Kruger effect, where individuals with lower competence tend to overestimate their abilities, and optimism bias, which leads to inflated expectations of positive outcomes and a disregard for potential risks Kruger & Dunning (1999); Sharot (2011).

In the context of reinforcement learning, particularly in $Q$-learning, a similar phenomenon arises. The overestimation of $Q$-values occurs because the algorithm maximizes over estimated rewards, which, when subject to noise or approximation errors, results in upwardly biased value predictions. This behavior parallels human decision-making, where overly optimistic assessments often lead to suboptimal choices in situations of uncertainty Kahneman (2011).

Mathematically, in $Q$-learning with discrete actions Watkins & Dayan (1992), the value update is performed using a greedy target, defined as:

$$y = r + \gamma \max_a Q(s, a) \tag{5}$$

Here, the greedy policy maximizes over action values, but if errors are present in the estimation of $Q(s, a)$, the expected value of this maximum is typically greater than the true value, as shown by Jensens inequality Jensen (1906):

$$\mathbb{E}_\epsilon \left[ \max_a (Q(s, a) + \epsilon) \right] \geq \max_a Q(s, a) \tag{6}$$

Even when errors are initially zero-mean, this mechanism results in a persistent overestimation bias, which is propagated through the Bellman updates. As errors are inherent in function approximation, this bias poses a significant challenge in practice An et al. (2021); Shi et al. (2022); Kakade & Langford (2002); Heger (1994); Kumar et al. (2020); Lyu et al. (2022).

On the other hand, TD3 Fujimoto et al. (2018) introduces a lower bound approximation by using two $Q$-networks to form a conservative $Q$-target. However, this approach has its limitations, as maximizing the lower bound can result in a policy that is overly concentrated near a specific maximum. When the critic's estimates are erroneous and the maximum is spurious, this can lead to a systematic underestimation bias, resulting in suboptimal performance Ciosek et al. (2019). Mathematically, the expected value under such conditions follows:

$$\mathbb{E}_\epsilon \left[ \min_{i=1...N} \max_a (Q(s, a) - \epsilon) \right] \leq \min_{i=1...N} \max_a Q(s, a) \tag{7}$$

This behavior underscores the inefficiency of relying solely on lower bound approximations for exploration.

While methods like Double $Q$-learning and ensemble techniques have been introduced to address overestimation bias in reinforcement learning, they often introduce a new issue: underestimation bias. This occurs as a result of overly conservative $Q$-value estimates, which can skew action selection towards suboptimal choices. In Double $Q$-learning, the decoupling of action selection from evaluation reduces the risk of overestimation, but it can also lead to underexploration, as the policy becomes too cautious in selecting actions. Similarly, ensemble methods that aggregate multiple $Q$-networks to mitigate overestimation may suffer from excessive pessimism, where the policy undervalues potential rewards. This underestimation bias stifles exploration, ultimately degrading performance in more complex tasks. Thus, while these methods provide a safeguard against overoptimism, their tendency to undervalue future rewards requires careful balancing to avoid underexploration and the associated performance decline.

## 4 STD $Q$-TARGET

Applying the SQT algorithm to any ensemble-based actor critic framework Konda & Tsitsiklis (1999), is straightforward and involves only minor adjustments. By integrating a few additional lines of code into the $Q$-target formula, one can compute the SQT values and subtract them from the $Q$-target estimations values.

The primary goal of SQT is to address the issue of overestimation bias, which can significantly impact an algorithm's performance. SQT aims to offer a minimal yet effective solution to this problem.

For an actor critic ensemble algorithm, the SQT's $Q$-target equation is:

$$y = r + \gamma \underbrace{\mathcal{Q}_{i=1...N}}_{\text{Ensemble } Q-\text{target operator}} [Q_i(s', a')] - \alpha \cdot \underbrace{\text{SQT}[\mathcal{B}]}_{\text{Ensemble uncertainty penalty}}, \quad a' \sim \pi(s) + \epsilon, \epsilon \sim \mathcal{U}(0, \sigma) \tag{8}$$

As in DPG-based algorithms Silver et al. (2014), SQT's actor is updated by the mean-batch of the mean $Q$-networks, with the following formula:

$$\nabla_{\theta^\mu} J \approx \mathbb{E}_{s_t \sim \rho^\beta}[\nabla_{\theta^\mu}(\underbrace{N^{-1} \sum_{i=1...N} Q_i(s,a|\theta^Q))}_{\text{Mean ensemble}}|_{s=s_t, a=\mu(s_t|\theta^\mu)}] \tag{9}$$

In summary, the SQT algorithm can be succinctly described as a **conservative $Q$-target formula** that subtracts a $Q$-function disagreement term from the algorithm's $Q$-values.

The SQT algorithm can be summarized by the following pseudo-code:

### 4.1 ALGORITHM

---
**Algorithm 1** SQT
---
1: **for** each iteration $t$ **do**
2:     Take a step $a$ in state $s$ using Equation:

$$\mu'(s_t) = \mu(s_t|\theta_t^\mu) + \underbrace{\mathcal{N}}_{\text{Random noise}} \tag{10}$$

3:     Store tuple $\mathcal{D} \leftarrow \mathcal{D} \cup \{(s,a,r,s',d)\}$.
4:     **for** each iteration $g \in G$ **do**
5:         Sample batch, $\mathcal{B} \sim \mathcal{D}$, using the algorithm's sampling rule.
6:         Compute $Q$-target $y$ using Equation 8.
7:         Update critic parameters using Equation 3.
8:         Update actor parameters using Equation 9.
9:         Update target parameters using Equation:

$$\theta' \leftarrow \theta \tag{11}$$

        at the specified interval.
10:    **end for**
11: **end for**

---

## 5 THEORETICAL PROPERTIES

**Theorem 5.1.** *In tabular settings SQT algorithm converges towards the optimal Q-values, when there are infinite updates for every state-action pair, within a finite MDP, assuming the step size of $\sum_t \alpha_t = \infty, \quad \sum_t \alpha_t^2 < \infty$ and $\gamma < 1$.*

$$\lim_{t=\infty}[\min_{i=1...N} Q(s,a) - SQT[\mathcal{B}]] = Q^*(s,a) \tag{12}$$

*Proof.* Tabular generalized $Q$-learning with a $Q$-values operator $\mathcal{G} : \mathbb{R}^{nNK} \to \mathbb{R}$ converges to the optimal $Q$-values $Q^*$ under the conditions of (by Lan et al. (2020)):

**Assumption 5.2.** *Let:*

    *1.*     $\underbrace{\mathcal{G}}_{\text{Ensemble } Q-\text{target operator}}$    $: \mathbb{R}^{nNK} \to \mathbb{R}$, and $\mathcal{G}(Q) = q \in \mathbb{R}$ (part 1)

    *2.* $Q = Q_{i,j,a} \in \mathbb{R}^{nNK}, a \in \mathcal{A}, |\mathcal{A}| = n, i \in \underbrace{\{1...N\}}_{\text{Ensemble } Q-\text{networks}}$ *, and* $j \in \underbrace{\{0...K-1\}}_{\text{Update time steps}}$ *(part 2)*

*Then:*

    *1. If $\forall i$ and $k$, $\forall j$ and $l$, and $\forall a \in \mathcal{A} : Q_{i,j,a} = Q_{k,l,a}$*

        *Then: $q = \max_a Q_{i,j,a}$*

2. $\forall \underbrace{Q}_{Q-networks\ ensemble\ 1} \in \mathbb{R}^{nNK}$ and $\underbrace{Q'}_{Q-networks\ ensemble\ 2} \in \mathbb{R}^{nNK}$:

$$|\mathcal{G}(Q) - \mathcal{G}(Q')| \leq \max_{a,i,j} |Q_{i,j,a} - Q'_{i,j,a}|$$

The key idea behind the proof is that, at convergence, the SQT penalty term becomes negligible, as both $Q$-networks tend to converge to the same point. As a result, the ensemble's $Q$-value update rule for SQT gradually reduces to that of the generalized MaxMin method Lan et al. (2020), thus satisfying all the conditions required for the tabular $Q$-learning convergence.

To formalize this, we start with the ensemble $Q$-target operator $\mathcal{G}$, which in the case of SQT is defined as:

$$\mathcal{G}^{MQ}(Q_s) = \max_{a \in \mathcal{A}} \min_{i \in \{1...N\}} Q_{i,s,a} - \text{SQT}[\mathcal{B}] \tag{13}$$

where the term $\text{SQT}[\mathcal{B}]$ represents the penalty applied to the ensemble. However, when all $Q$-networks converge, i.e., $Q_{i,j,a} = Q_{k,l,a}$ for all $i, j, k, l$, and actions $a$, the penalty term diminishes:

$$\text{SQT}[\mathcal{B}] = 0 \tag{14}$$

This leads to the simplification of the update rule to the form used in MaxMin's formula:

$$\mathcal{G}^{MQ}(Q_s) = \max_{a \in \mathcal{A}} \min_{i \in \{1...N\}} Q_{i,s,a} \tag{15}$$

At convergence, this matches the MaxMin operator, fulfilling part 1 of the conditions required for convergence. Moreover, since the SQT term vanishes, the operator becomes identical to the generalized MaxMin operator, which ensures that the optimal $Q$-values are reached as demonstrated in Lan et al. (2020).

Finally, part 2 of the convergence condition is also satisfied because the difference between the $Q$-values for any state-action pairs in different $Q$-networks becomes bounded by their maximum deviation:

$$|\mathcal{G}_{\text{SQT}}(Q_s) - \mathcal{G}_{\text{SQT}}(Q'_s)| \leq \max_{a,i} |Q_{i,s,a} - Q'_{i,s,a}| \tag{16}$$

Thus, SQT, at convergence, adheres to the same convergence properties as MaxMin's generalized $Q$-learning framework, with the added benefit of more robust $Q$-estimates during training due to the penalty term before convergence.

$\square$

# 6 EVALUATION

We evaluated the SQT algorithm[1] across a range of simulated locomotion tasks that varied in complexity. These tasks were implemented in the MuJoCo simulator Todorov et al. (2012) using Gym Brockman (2016), and involved applying torques to actuated joints for movement control. The environments tested included cheetah, walker, ant, and humanoid, with their corresponding state and action space dimensions outlined in table 1. The hyperparameters employed in the experiments were set to $\gamma = 0.99$ and $\alpha_{\text{sac}} = 0.2$, with $\alpha \in (0, 1]$ fine-tuned according to the specific algorithm and task at hand.

To ensure a robust comparison, we evaluated SQT against TD3 Fujimoto et al. (2018), SAC Haarnoja et al. (2018), and TD7 Fujimoto et al. (2023), the latter serving as the base algorithm upon which

---

[1]SQT code: https://github.com/anonymouszxcv16/SQT

SQT was implemented. Each algorithm was tested across five random seeds, $\{0, 1, 2, 3, 4\}$, and performance was measured by the maximum average reward obtained over 1,000,000 steps, averaged across the different seeds.

| Task | State Dimension | Action Dimension |
|---|---|---|
| Cheetah | 17 | 6 |
| Walker | 17 | 6 |
| Ant | 27 | 8 |
| Humanoid | 376 | 17 |

Table 1: Summary of task characteristics, highlighting state and action space dimensions.

The table summarizes the key properties of the MuJoCo tasks used for testing, highlighting their state and action space dimensions. These tasks vary in complexity, with cheetah and walker representing simpler environments, both having a state dimension of 17 and an action dimension of 6. The ant task, with higher complexity, features a state dimension of 27 and an action dimension of 8. The most complex task, humanoid, has a significantly larger state dimension of 376 and an action dimension of 17, reflecting its greater task difficulty and diversity of control challenges.

The results are summarized in the following table 2.

| Environment | TD7 | SQT | TD3 | SQT | SAC | SQT |
|---|---|---|---|---|---|---|
| Cheetah | 17.1K | **17.7K** | 9.6K | **9.7K** | 2.6K | **4.7K** |
| Humanoid | 6.7K | **8.1K** | 5.0K | **6.5K** | 3.2K | **5.7K** |
| Walker | 6.0K | **7.1K** | 5.1K | **5.3K** | 2.7K | **2.8K** |
| Ant | 8.3K | **8.9K** | 3.8K | **5.9K** | 929.4 | 929.4 |
| **Improvement** | | **+12.15%** | | **+21.6%** | | **+34.04%** |

Table 2: Comparison of SQT with TD7, TD3, and SAC across various MuJoCo tasks.

The results presented in table 2 demonstrate a clear performance advantage for SQT when applied to SOTA actor-critic algorithms across a range of MuJoCo tasks. Notably, SQT consistently outperforms its base algorithms–TD7, TD3, and SAC–showing significant improvements, particularly on more complex tasks like humanoid and ant. This indicates the efficacy of SQT in enhancing $Q$-value estimates, which leads to more effective policy learning.

A key insight from the results is the **correlation between the optimism of the base algorithm and the magnitude of improvement** when enhanced with SQT. SAC, known for being the most optimistic due to its entropy-regularized actor, sees the highest improvement with SQT, boasting a remarkable 34.04% performance increase. TD3, which is somewhat optimistic but less so than SAC, achieves a medium improvement of 21.6%. Finally, TD7, which is the most conservative of the three due to its reliance on dynamics learning and a high UTD ratio (forcing the policy to closely follow its own learned actions), experiences the lowest improvement at 12.15%.

This pattern suggests that SQT excels particularly well when paired with more optimistic algorithms. The significant boost in performance for SAC, which relies on stochastic policies and exploration, shows that SQT helps mitigate overestimation errors in optimistic settings, resulting in more accurate $Q$-value estimations and superior learning efficiency. On the other hand, for algorithms like TD7, which prioritize stability and conservative policy updates, the impact of SQT is still positive but more muted, as the algorithm already enforces a strong regularization in its learning dynamics.

In complex environments like humanoid and ant, where high-dimensional state-action spaces make overestimation errors more costly, SQT's improvements are even more pronounced. For example, in humanoid, SQT boosts TD3's performance by 30% (from 5.0K to 6.5K) and SAC's by 78% (from 3.2K to 5.7K), demonstrating its ability to handle challenging tasks effectively.

Overall, the results clearly highlight SQT as a robust enhancement to actor-critic algorithms, with particularly strong performance gains in environments and algorithms where optimism plays a key role.

## 7 EXPERIMENTAL ANALYSIS

To evaluate the diversity in $Q$-network estimations throughout the learning process, we conducted a comparison of the standard deviations of the $Q$-networks between TD3 and SQT across different environments. Table 3 summarizes the results, showing the average batch $Q$-network standard deviations, computed as the mean of evaluation intervals of 5,000 timesteps. Each interval represents the average batch standard deviation of the $Q$-networks, averaged over 1,000,000 steps and averaged across five random seeds.

The standard deviation over an evaluation interval, denoted as $\text{std}_T$, is defined by the following equation:

$$\text{std}_T = \frac{1}{T} \sum_{0 \ldots T} \text{std}[\mathcal{B}_t] \tag{17}$$

where $T = 5,000$ is the interval length, and $\mathcal{B}_t \sim \mathcal{D}$ represents a sampled batch at timestep $t$. The standard deviation of a batch, $\text{std}[\mathcal{B}]$, is calculated as:

$$\text{std}[\mathcal{B}] \leftarrow \frac{1}{|\mathcal{B}|} \sum_{(s,a) \sim \mathcal{B}} \text{std}_{i=1,2} Q_i(s,a) \tag{18}$$

This formulation captures the diversity in $Q$-network predictions by averaging the individual $Q$-network standard deviations for each state-action pair within the batch.

| Environment | TD3 | SQT |
|:---:|:---:|:---:|
| Cheetah | 1.04 | **1.17** |
| Humanoid | 0.93 | **2.11** |
| Walker | **1.50** | 1.24 |
| Ant | 1.37 | **1.65** |
| **Improvement** | | **+35.4%** |

Table 3: $Q$-networks std for TD3 vs. TD3 + SQT. Higher values indicate more diverse ensemble $Q$-networks estimates.

The analysis of the results shows that SQT consistently achieves higher $Q$-network standard deviations across most environments, highlighting its ability to foster more diverse $Q$-value estimations. This advantage in standard deviation directly translates to improved exploration capabilities, which is critical in reinforcement learning. Notably, in environments such as cheetah, humanoid, and ant, SQT significantly outperforms TD3, with the humanoid task demonstrating the largest difference. Here, SQT achieves a $Q$-network standard deviation of 2.11 compared to TD3's 0.93, which correlates with the increased complexity of the humanoid environment. This greater diversity suggests that SQT introduces more variability in the ensemble predictions, which in turn enhances exploration and reduces overestimation bias, leading to more robust policy learning.

In the walker environment, TD3 exhibits a slightly higher $Q$-network standard deviation than SQT. Nevertheless, the overall performance of SQT in terms of $Q$-value diversity remains impressive, with an average improvement of +35.4% across all environments. This superior diversity is a key factor in SQT's ability to improve exploration, making it particularly effective in handling complex tasks where diverse value estimates are essential for exploring high-dimensional state-action spaces and making more reliable predictions.

## 8 ABLATION STUDY

In this ablation study, we investigate the effect of using a state-action-specific penalty for the $Q$-networks' standard deviation compared to the batch mean penalty, which is the default in SQT. The goal is to analyze the performance implications of incorporating more localized uncertainty estimates (i.e., per state-action pair) versus a more global penalty (i.e., averaged over the entire

batch). By isolating these two approaches, we aim to better understand the trade-offs between fine-grained uncertainty penalization and the computational simplicity of a single, aggregate penalty.

In the tuple-based SQT, we calculate the standard deviation of the $Q$-network outputs for each individual state-action pair in the batch, and directly incorporate this value into the $Q$-target update for that pair. The updated $Q$-target formula becomes:

$$Q_{\text{SQT}_{\text{tuple}}}(s, a) = r + \gamma \left( \mathcal{Q}_{i=1,2}[Q](s', a') - \alpha \cdot \text{std}_{i=1,2} Q_i(s', a') \right), \tag{19}$$

where $(s, a) \in \mathcal{B}$ represents a state-action pair from a sampled batch $\mathcal{B} \sim \mathcal{D}$, $a' \leftarrow a'_{\text{pure}} + \epsilon$ is the action perturbed by noise, with $a'_{\text{pure}} \sim \pi_\theta(s')$ being the action sampled from the policy, $\epsilon \sim \mathcal{U}(0, \sigma)$ denotes uniformly sampled noise, and $\alpha \in (0, 1]$ is a parameter that regulates the magnitude of the penalty.

In the default SQT formulation, the standard deviation is computed across the entire batch of state-action pairs, and this single value is applied as a penalty for all the updates in the batch:

$$Q_{\text{SQT}}(s, a) = r + \gamma \left( \mathcal{Q}_{i=1,2}[Q](s', a') - \alpha \cdot \text{mean}_{(s',a') \sim \mathcal{B}}[\text{std}_{i=1,2} Q_i(s', a')] \right), \tag{20}$$

To assess the impact of the two penalty formulations, we conducted experiments on the same set of MuJoCo locomotion tasks as in the primary experiments (cheetah, walker, ant, and humanoid), using SQT-based algorithms built on top of TD3. The hyperparameters were kept consistent with the main experiments, with $\alpha_{\text{tuple}}$ set to 0.1.

Table 4 presents the performance comparison between the two variants. The key metric is the maximum average reward over 1,000,000 steps, averaged across five random seeds.

| Environment | SQT | SQT$_{\text{tuple}}$ |
|---|---|---|
| Cheetah | 9.7K | **10.2K** |
| Humanoid | **6.5K** | 2.8K |
| Walker | 5.3K | **5.5K** |
| Ant | 5.9K | **6.5K** |
| **Improvement** | | -9.0% |

Table 4: A comparative evaluation was performed between SQT and SQT$_{\text{tuple}}$, both implemented on top of TD3, to analyze their performance differences.

The table results illustrate a clear distinction between the performance of the standard SQT and SQT$_{\text{tuple}}$, particularly across environments of varying complexity. In simpler tasks such as cheetah, walker, and ant, SQT$_{\text{tuple}}$ outperforms SQT by applying a local penalty based on the standard deviation for each state-action pair. This localized penalty approach appears more effective in these lower-dimensional environments, where the simplicity allows for precise adjustments, leading to better performance, as seen in the 10.2K, 5.5K, and 6.5K reward scores respectively.

However, in the most complex environment, humanoid, SQT demonstrates clear superiority, achieving a significantly higher score of 6.5K compared to SQT$_{\text{tuple}}$'s 2.8K. This highlights the advantage of SQT's global penalty mechanism, which leverages the wisdom of the crowd principle by applying a single penalty based on the batch standard deviation across all state-action pairs. In more complex environments like humanoid, where large state and action spaces introduce greater uncertainty, this global penalty helps smooth out inconsistencies by considering the broader context of the entire batch. This approach makes SQT more robust in high-dimensional settings, effectively reducing the overestimation bias and leading to more reliable policy improvements.

While SQT$_{\text{tuple}}$ excels in simpler environments due to its precise, localized penalties, the global batch penalty in SQT proves essential for solving complex tasks, as evidenced by its superior performance in the humanoid environment. Overall, this demonstrates the strength of SQT in handling greater uncertainty in complex environments by utilizing the ensemble of predictions across the batch to balance exploration and exploitation.

## 9 RELATED WORK

Overestimation bias is a common challenge in reinforcement learning, particularly in value-based methods that rely on function approximators like deep neural networks. Several approaches have been proposed to mitigate this issue, each with its own strengths and limitations.

Double $Q$-learning Hasselt (2010) was one of the earliest approaches designed to counter overestimation bias. It achieves this by employing two distinct $Q$-tables, decoupling the processes of action selection and action evaluation. Specifically, the action chosen by one $Q$-table, $Q^A$, is used to update the other $Q$-table, $Q^B$, and vice versa. This separation reduces the likelihood of both $Q$-tables overestimating the same action value. Although Double $Q$-learning effectively addresses overestimation, its applicability is limited to tabular environments and does not scale well to continuous action spaces or high-dimensional problems. A variant, Cini et al. (2020), extends this framework.

TD3 Fujimoto et al. (2018) adapts the Double $Q$-learning principle to actor-critic architectures by utilizing two $Q$-networks to compute a conservative $Q$-target. The approach mitigates overestimation bias by selecting the smaller value between the two critics when determining the $Q$-target, thereby minimizing the risk of optimistic estimates. Despite its improvements over standard actor-critic techniques, TD3 can become excessively conservative in cases where overestimation bias is less significant, potentially hindering learning speed and exploration efficiency. Variants such as Fujimoto & Gu (2021); Fujimoto et al. (2019); Chen et al. (2021) build upon the TD3 framework.

SAC Haarnoja et al. (2018) tackles overestimation bias by employing a maximum entropy framework, which enhances exploration by incorporating an entropy term into the $Q$-target. Specifically, this method subtracts the log probability of the policy from the $Q$-target, encouraging more exploratory actions and preventing the policy from becoming overly deterministic. While SAC effectively reduces overestimation bias and promotes exploration, it may result in suboptimal performance in environments where extensive exploration is unnecessary. Additionally, the algorithm can become computationally expensive due to the inherent stochasticity in policy updates. Several variants, such as D'Oro et al. (2022); Ciosek et al. (2019), build on the SAC framework.

Bootstrapped $Q$-networks Osband et al. (2016) use an ensemble of $Q$-networks to generate diverse estimates of the $Q$-values, encouraging a wider exploration of possible action values. Each member of the ensemble is trained with different initial conditions or data subsets. By taking the minimum $Q$-value across the ensemble, this method can mitigate overestimation bias while maintaining sufficient exploration. However, Bootstrapped $Q$-networks are computationally expensive due to the overhead of maintaining and training multiple critics, and the methods effectiveness can diminish if the ensemble members fail to remain sufficiently diverse.

MaxMin $Q$-learning Lan et al. (2020) extends TD3 by employing an ensemble of $N$ critics rather than two, balancing between optimism and pessimism based on the size of the ensemble. The $Q$-target is the minimum of a subset of $K$ critics from the ensemble, rather than the full ensemble, to avoid excessive conservatism. While MaxMin $Q$-learning mitigates overestimation more effectively than TD3, it introduces a trade-off between computational efficiency and conservatism. Larger ensembles offer better bias control but at the cost of greater computation, while smaller ensembles may not fully suppress overestimation bias.

TD7 Fujimoto et al. (2023) is a recent method that enhances TD3 by incorporating State Action Learned Embedding (SALE) into the critic networks. SALE models the transition dynamics and augments the input space of the critics and the policy, improving the accuracy of value estimates. TD7 addresses overestimation bias similarly to TD3 by using multiple critics but also incorporates learned representations of the environment's dynamics to refine the $Q$-value predictions. However, the additional complexity introduced by SALE can increase the difficulty of training and may require more extensive tuning for different environments.

SQT presents a notable advantage over traditional methods by integrating an ensemble-based disagreement penalty, which is rooted in the well-established mathematical concept of standard deviation (std) Pearson (1920) for quantifying uncertainty. This strategy helps maintain conservativeness in $Q$-values, effectively minimizing overestimation bias while also fostering exploration. The penalty mechanism utilized in SQT takes advantage of the diversity among $Q$-networks, motivating them to learn distinct features by differentiating their predictions from each other. Consequently,

this promotes exploration, allowing the $Q$-networks to capture various aspects of the environment's knowledge instead of rapidly converging to similar solutions.

In contrast, other methods, such as Double $Q$-learning or ensemble-based conservative formulas, prioritize conservativeness without considering exploration. These approaches, which rely on minimal $Q$-network ensembles or the double estimator, tend to suffer from underexploration due to their overly pessimistic nature. The excessive focus on conservativeness, although useful for bias reduction, ultimately leads to poor performance. This is particularly evident in tasks requiring greater exploration, where the lack of sufficient exploratory mechanisms restricts the algorithm's ability to discover better strategies, despite considerable computational effort.

SQT overcomes this by balancing conservativeness and exploration, maintaining safe $Q$-values while encouraging sufficient exploration, thus offering a more robust solution that avoids the pitfalls of underexploration found in other conservative methods. This balance allows for improved performance across a wider range of environments with reduced computational overhead.

## 10 CONCLUSION

SQT represents a novel $Q$-learning model-free online on-policy actor critic algorithm, specifically designed to address the overestimation bias present in RL algorithms. The central innovation of SQT is the inclusion of an uncertainty penalty within the $Q$-target formula, which is determined by the disagreement among $Q$-networks.

We integrated SQT with TD3, SAC, and TD7, resulting in an ensemble-based conservative actor-critic approach. The performance of SQT was evaluated against SOTA actor critic model-free online algorithms–across widely-used MuJoCo locomotion tasks: humanoid, walker, ant, and cheetah. The results consistently showed significant performance improvements with SQT, surpassing TD3, SAC, and TD7 on the majority of tested tasks.

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
