# OpenReview forum: "SQT -- rough conservative actor critic"
_ICLR.cc/2025/Conference — Submitted to ICLR 2025_

### Official Review · Reviewer_epyW · 2024-10-27

**Soundness:** 2
**Presentation:** 1
**Contribution:** 2
**Rating:** 3
**Confidence:** 2

**Summary:**

Authors propose a regularization method for conservative Q-function training by using the uncertainty of the critic networks. Empirical results demonstrate the performance improvement on average.

**Strengths:**

1. Simple nature of modification
2. Some theoretical analysis
3. Improved empirical performance

**Weaknesses:**

1. I didn't like the way the paper is structured. The method is described in section 8 while most of the results are presented  before. I could not understand what is the proposed approach until the page 8. I strongly recommend authors to put the description of their approach (what is SQT operator) in section 4.
2. Unclear motivation and description of the main approach. In equation 20 authors demonstrate the formula for their main approach, if I understand correctly. However what are s' and a' under the **mean** part? Those are supposed to be the next state and action in the other part of equation. Is it just a random set of states and actions from the replay buffer? If so, I do not see the motivation behind the schema. I fish authors clarify and motivate it better. I.e. why random set of state-actions is useful for learning Q(s, a) for a particular (s, a) pair? In case of SQT_{tuple} I have no questions as we use only the following pair.
3. Questionable claim about the superiority of SQT when compared to SQT_{tuple}. It outperforms only on one task out of four. It looks very suspicious and the claim " This highlights the advantage of SQT’s global penalty mechanism, which leverages the wisdom of the crowd principle by applying a single penalty based on the batch standard deviation across all state-action pairs" seems too strong to me. I consider it is as critical to show on more tasks, otherwise the efficiency is very questionable.
4. Limited evaluation. I'm not strongly aware about environments which are used for online RL approaches now but at least there should be a Hopper MuJoCo environment. Are there any more environments where authors can demonstrate that SQT is better than SQT_{tuple}?
5. No statistical significance is reported. There are only mean scores in the tables and no learning curves.
6. No offline RL results which could be very relevant here.

**Questions:**

Most of my questions are based on Weaknesses section
1. Please report the Hopper environment scores and if possible some other environments to support your claims.
2. It would be very nice to see offline RL performance using at least the subset of D4RL tasks that was reported in TD7.
3. Please report stds and learning curves.
4. I'm also confused about some of the theoretical propositions. In https://arxiv.org/abs/2110.01548 authors claim that under some assumptions taking the min over critics is equivalent to the subtracting critics std with some coefficient. From your work this claim seems to be wrong. Could you elaborate on it please?
5. What is the impact of hyperparameters that are used by your approach, i.e. batch size and $\alpha$?

---

### Official Review · Reviewer_dzex · 2024-10-28

**Soundness:** 2
**Presentation:** 1
**Contribution:** 2
**Rating:** 1
**Confidence:** 5

**Summary:**

This paper proposes a conservative actor-critic algorithm named SQT (Standard Q-target). It uses the combination of predictions from N distinct Q-networks and their standard deviation to compute the Q-target value and uses the mean of Q networks’ values to update the actor-network. It implements the method on TD7/TD3/SAC algorithms, improving their performance.

**Strengths:**

1.	This paper improves the performance of the TD7/TD3/SAC algorithms through its SQT method.

**Weaknesses:**

1. The contributions of this paper are insufficient. The primary contribution is simply the addition of the standard deviation of ensemble networks to the Q-target values. The authors should consider further how the experimental effect differs for different penalty terms, or give an explanation for why the standard deviation is used as a penalty term, and not some other difference, e.g. variance. In addition, the authors need to conduct ablation experiments on the hyperparameter in front of the penalty term to study the effects of different hyperparameters on the test results.
2. The use of ensemble Q estimators is not particularly novel and the authors did not cite the relevant papers in Background section, such as some related ensemble learning methods.
3. The citation format in the paper is inconsistent, and the use of abbreviations and table formatting are irregular, affecting the paper's readability and adherence to standards. Such as Eq. (10) first appears in the pseudocode without any explanation in the text. It is recommended to provide an explanation when the formula is introduced for the first time. Additionally, when abbreviations such as 'RL' and 'MDP' appear for the first time, the full terms also should be provided.
4. The experiments in the paper only compare a few benchmark algorithms and do not include more recent reinforcement learning algorithms. It is recommended to include additional benchmark comparisons and statistical analyses, such as DreamerV2 and REM.

**Questions:**

1.	Why did you choose the standard deviation as the penalty term, rather than other measures of uncertainty?
2.	What is the optimal value for the $\alpha$ parameter in front of the standard deviation, and have you experimented with it?
3.	Why did you select these specific algorithms for comparison? Have you considered comparing them with more recent reinforcement learning algorithms?

---

### Official Review · Reviewer_j4AC · 2024-11-04

**Soundness:** 2
**Presentation:** 3
**Contribution:** 3
**Rating:** 3
**Confidence:** 4

**Summary:**

This paper proposes Std Q-target, which uses disagreement in Q ensemble to get an uncertainty penalty, giving a minimalist solution to Q overestimation bias. The authors applied the proposed approach to SAC, TD3 and TD7 and show consistent performance gain on MuJoCo benchmark.

**Strengths:**

**originality**
- Although the proposed approach is very simple, the authors show it can lead to strong performance improvement when used properly

**quality**
- overall good quality
- Simplicity: I appreciate the simplicity of the proposed approach.

**clarity**
- Overall paper is clear, easy to follow

**significance**
- strong empirical results
- Table 2 shows for the three baseline algorithms, adding SQT will help performance consistently.

**Weaknesses:**

- Most important: Lack of in-depth analysis of the effect of the proposed approach on bias reduction. The only thing shown in the paper is the performance, give the proposed change is quite simple, the paper can benefit from more related experiments to help understand the issue better. For example, one way to do this is to show how the amount of bias changes for different algorithms, you can get the Q predictions and estimate the true Q values with MC return (as done in the REDQ paper you cited), or you can look at how Double Q learning paper ("Deep Reinforcement Learning with Double Q-learning") did it.
- Lack of technical details: there is no adequate discussion of hyperparameters and hyperparameter sensitivity, and other details such as how much computation the method requires compared to the baselines?
- There are other algorithms that utilize the uncertainty of Q ensembles to help reduce bias, such as TQC ("Controlling Overestimation Bias with Truncated Mixture of Continuous Distributional Quantile Critics") and REDQ ("Randomized Ensembled Double Q-Learning: Learning Fast Without a Model"), how does the proposed approach compare to these methods?
- Since this paper focuses on empirical results, the paper can be made stronger by providing more extensive results, such as comparing to more baselines and test on more benchmarks (e.g. other MuJoCo environments, deep mind control, Atari).

Minor issues:
- Line 309 you may want to introduce what UTD stands for here.

**Questions:**

- How do we know that the proposed approach indeed improves performance by mitigating the bias issue?

---

### Official Review · Reviewer_ZdLV · 2024-11-04

**Soundness:** 1
**Presentation:** 2
**Contribution:** 1
**Rating:** 1
**Confidence:** 4

**Summary:**

This paper considers online RL: substact the std of ensemble Q values from the value backup. Experiments done on 4 MuJoCo tasks.

**Strengths:**

- The paper is relatively clear to understand.

**Weaknesses:**

- This exact idea has already been published before: *"Why So Pessimistic? Estimating Uncertainties for Offline RL through Ensembles, and Why Their Independence Matters"* (Ghasemipour et al, NeurIPS 2022). See its *Shared Targets (Method 2)*.
- Experimental results are extremely weak.

**Questions:**

See Weakness Section.

---

### Meta-Review · Area_Chair_ZBNq · 2024-12-23

**Metareview:**

This paper proposes SQT (Standard Q-target), a conservative actor-critic algorithm that uses an ensemble of Q-networks. It calculates the Q-target using the mean and standard deviation of predictions from multiple Q-networks and updates the actor-network using the mean.

The major issues raised by the reviewers lie in following aspects:

1, Lack of Novelty: Using ensemble Q estimators is not new, and relevant prior work is not cited.

2, Limited Experiments Evaluation: The experiment results in the paper is weak and with limited baselines and testing benches.

3, Lack of technical details and in-depth analysis.

**Additional Comments On Reviewer Discussion:**

The authors gave up rebuttal. All the reviewers achieve the agreement on this paper.

---

### Decision · Program_Chairs · 2025-01-22

Reject